# CView: A network based tool for enhanced alignment visualization

Raquel Linheiro[1☯], Stephen Sabatino[1,2☯], Diana Lobo[1,2,3☯], John Archer[1,2]*

**1** CIBIO, Centro de Investigação em Biodiversidade e Recursos Genéticos, *InBIO* Laboratório Associado, Campus de Vairão, Universidade do Porto, Vairão, Portugal, **2** BIOPOLIS, Program in Genomics, Biodiversity and Land Planning, CIBIO, Campus de Vairão, Vairão, Portugal, **3** Departamento de Biologia, Faculdade de Ciências, Universidade do Porto, Porto, Portugal

☯ These authors contributed equally to this work.
* john.archer@cibio.up.pt

**Data Availability Statement:** All code and test data is available from: https://sourceforge.net/projects/cview/ The data set described within the use case scenario is publicly available from the zenodo repository (https://zenodo.org/record/6475666),

## Abstract

To date basic visualization of sequence alignments have largely focused on displaying per-site columns of nucleotide, or amino acid, residues along with associated frequency summarizations. The persistence of this tendency to the recent tools designed for viewing mapped read data indicates that such a perspective not only provides a reliable visualization of per-site alterations, but also offers implicit reassurance to the end-user in relation to data accessibility. However, the initial insight gained is limited, something that is especially true when viewing alignments consisting of many sequences representing differing factors such as location, date and subtype. A basic alignment viewer can have potential to increase initial insight through visual enhancement, whilst not delving into the realms of complex sequence analysis. We present CView, a visualizer that expands on the per-site representation of residues through the incorporation of a dynamic network that is based on the summarization of diversity present across different regions of the alignment. Within the network, nodes are based on the clustering of sequence fragments that span windows placed consecutively along the alignment. Edges are placed between nodes of neighbouring windows where they share sequence identification(s), i.e. different regions of the same sequence(s). Thus, if a node is selected on the network, then the relationship that sequences passing through that node have to other regions of diversity within the alignment can be observed through path tracing. In addition to augmenting visual insight, CView provides export features including variant summarization, per-site residue and kmer frequencies, consensus sequence, alignment dissection as well as clustering; each useful across a range of research areas. The software has been designed to be user friendly, intuitive and interactive. It is open source and an executable jar, source code, quick start, usage tutorial and test data are available (under the GNU General Public License) from https://sourceforge.net/projects/cview/.

## Introduction

Tools developed to visualize local sets of aligned sequences, such as those produced by multiple sequence aligners including MUSCLE [1] and Clustal W [2], have focused largely on

under the permanent DOI 10.5281/zenodo.6475666 as is described in the manuscript.

**Funding:** This work was funded by the project NORTE-01-0246-FEDER-000063, supported by Norte Portugal Regional Operational Programme (NORTE2020), under the PORTUGAL 2020 Partnership Agreement, through the European Regional Development Fund (ERDF), and by research funding from the project under the references PTDC/BIA-EVF/29115/2017 and POCI-01-0145-FEDER-029115 co-funded by Operational Competitiveness and Internationalization Program, Portugal 2020 and the European Union via the European Regional Development Fund (ERDF), and by National Funds through FCT. DL was supported by FCT through a PhD grant with the reference PD/BD/132403/2017. The funders had no role in study design, data collection and analysis, decision to publish, or preparation of the manuscript. FCT URL: https://www.fct.pt/.

**Competing interests:** The authors have declared that no competing interests exist.

displaying columns of nucleotide or amino acid characters, and highlighting the differences between such characters primarily through the use of colour [3–7]. More advanced sequence management and analysis packages, such as Geneious [8] and Mega [9], as well as the more recent tools designed for basic visualization of mapped read data, including IGV [10], GenomeView [11] and Tablet [12], incorporate a wide array of analysis, summarization and annotation options, but in terms of basic visualization they follow a similar approach. In light of the emergence of the vast quantities of sequence data generated during the last decades [13], alignment free approaches to sequence summarization have also proliferated [14, 15]. However, the per-site information associated with multiple sequence alignments is at times integral to many topics of bioinformatics research ranging from phylogenetics [16–18], co-evolution [19], recombination detection [20] and protein-protein interactions [21] to inter-species evolutionary dynamics, such as those involving pathogens [22–25] as well as predator-pray [26] relationships; to mention just a few. The direct observation of aligned residues, as well as the general per-site based summarization, not only provides an accessible view of per-site alterations between sequences within the alignment but also implicitly gives the end-user a level of reassurance in relation to data. However, initial insight gained about the overall alignment is limited, especially when viewing alignments consisting of many sequences representing varying factors of interest such as geographical location, subtype, treatment strategy, compartment and date (or time-point). Basic alignment visualization should have the potential to increase the level of initial insight within sequence datasets whilst not delving into the realms of more complex sequence analysis. Here we present CView, a simple multiple sequence alignment visualizer that incorporates a dynamic network that is based on a summarization of the diversity across different regions of the alignment. The immediate coupling of aligned sequences to such a network provides a way of visually tracking the context of observed diversity within characters that are currently onscreen to that of the surrounding regions of the alignment not currently in view. This provides the user with an increased intuitive and visual summarization of the context of this diversity.

CView provides a range of export features that can be applied to the entire alignment, to a specified region of the alignment, or to a specified region in conjunction with a specified subset of sequences. Such export features include variant summarization, per-site residue and kmer frequency matrixes, clustering, pairwise-distance matrixes as well as consensus sequence generation. For example, when the "Variant Frequencies" option of the "Title Search" menu is selected, a list of variants spanning the user-specified region of the alignment, from sequences containing a user-specified search criteria within the sequence title (such as a specific year), is created. This is done by identifying all unique residue permutations between the specified co-ordinates from the subset of sequences matching the search criteria and associating each with their frequency of occurrence. The titles of each sequence associated with individual variants are then outputted in conjunction with a summarization, relative to the most frequent variant, of the subsection of the alignment used. On the other hand if the "Variant Frequency" option of the "All Sequences" menu is selected variants will be identified from all sequences within the alignment instead of a subset matching a user search criteria. Such a feature has use in the tracking of viral populations, for example in searching for the presence of genotypic alterations such as those associated with immune escape [27], drug resistance [28], or co-receptor usage [29, 30]. Additionally, this feature has use in both clinical [31] and environmental [32–34] metagenomics, where the summarization of populations of microbes is of interest.

Other output features within CView function in a similar manner to that of variant summarization and an overview of each is provided within the user manual, available on the CView SourceForge wiki page, as well as within the in-software "Help" menu. A usage demonstration video [35], as well as a brief tutorial on identifying variants [36], has been provided on zenodo

repository, and are also accessible through the CView SourceForge project wiki page. Aside from features related to the extraction of secondary information from the alignment, CView provides the ability to dissect the alignment into subsets of sequences and regions; a task that is often laborious in the absence of a background in script development. For example, a user can export a specified region of sequences associated with a specific time-point, geographical location or body compartment, as long as the sequence titles have been labelled with such information. Such labelling is often as standard output feature of sequence repositories, for example, in the Los Alamos HIV sequence database the user can select options such as subtype, patient code, country and year to be included within the title of each sequence [37]. Such information may also be part of experimental design where information on compartment [38] or time-point [39] may be available.

Within CView the network, displayed directly below the alignment, is based on the clustering of sequences within windows placed consecutively across the alignment, where each cluster becomes a node. Within any given window if multiple nodes exist, each represents a different portion of the diversity present at that windows location. Edges are placed between nodes of neighbouring windows where they share differing regions of the same sequence(s). Thus, if a single node within a region of the alignment displaying multiple nodes is selected, then the relationship that all sequences passing through that node have to nodes within other regions of the alignment can be instantly observed by highlighting the paths through the rest of the network that the sequences within the selected node take. Here we describe how these networks are constructed and displayed. The clustering threshold used during network construction, as well as the number and width of windows, are specified on the user-interface through a series of user-friendly slider bars. Alterations are updated in real time, which allows the user to visually explore the variation present across the alignment. A parameter reset button allows the user to instantly reset all network parameter slider bars to default values. The software has been designed to be user friendly, intuitive and interactive and it, along with source code, a quick start guide and test data, is available (under the GNU General Public License) through the SourceForge project page https://sourceforge.net/projects/cview/.

## Methods

### Implementation

The interface has been designed for simplicity and clarity. It consists of four general areas of user-interaction (Fig 1) which are: (1) sequence view, (2) network view, (3) navigation and control and, (4) menu driven outputs. Within the software a brief graphical overview of these areas is available though the "Help" -> "Interface" menu option, as well as through a help button within the "navigation and control" area. Hint text is also displayed within the control panel area once various components of the interface are clicked on. A video tutorial located on the zenodo repository provides a general overview of the interface [35].

**(1) Sequence view.** Sequences are displayed above the alignment location indicator. The dynamic yellow bar associated with the latter represents the region of the alignment that is currently visible. The green dot on the right hand side indicates what proportion of the sequences are currently visible. The consensus sequence of the alignment is displayed along the top of the sequence area, and directly under this the "+" indicates columns where all characters agree with the consensus character. Sequences and their titles are clickable and when an individual sequence is selected it will be traced through the corresponding network as a yellow line. If a user clicks on a node within the network area, all sequences that pass through that node will have a red dot placed next to their titles. These red dots correspond to the red paths that will become visible on the network; the latter indicative of through which other nodes the

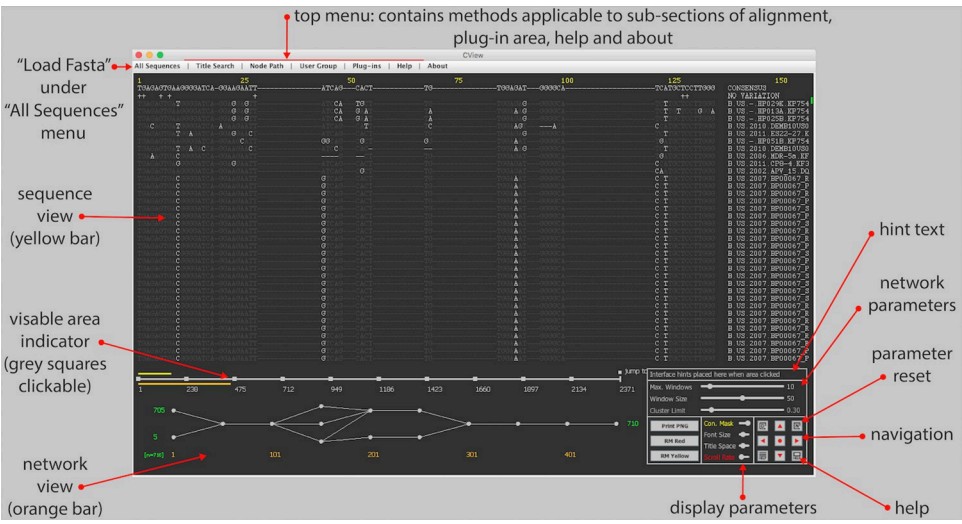

**Fig 1. CView interface.** The four main areas of the CView interface are depicted. These are sequence view, network view, control panel and the top menu. The yellow numbers on the top indicate the sites of the alignment that are currently in view. These correspond to the yellow bar on the top of the location indicator. The orange numbers along the bottom indicate the locations of the windows that nodes within the network are dependent on. These window locations correspond to the area that the orange bar located the under the location indicator covers. Grey dots indicate selectable nodes within windows. The squares along the location indicator can also be selected in order to jump directly to the indicated co-ordinates. The red text around the outside of the interface summarizes the main features.

sequences pass. Sequences possessing a red dot next to their titles in this manner will be placed on top of the sequence display list, i.e. the order of sequences is sorted with these on top. If a different node is subsequently selected, the red title dots are reallocated to sequences passing through the newly selected node, paths redrawn and the sequence list is resorted. Basic parameters associated with the sequence display include the masking of characters that are the same as consensus, thus making it easier to identify non-consensus residues by eye, altering font size, altering space allocated to display titles and scroll speed; these are achieved through the "Navigation and Control" panel that is located on the bottom right of the display. A button within the control panel area can be used to reset all display parameters to default values. Site locations are highlighted in yellow along the top of the interface.

**(2) Network view.** The network depicting sequence diversity within the alignment is displayed directly below the alignment location indicator. The associated orange bar of the latter represents the region of the alignment that is currently represented by the network. The region begins from the current sequence view starting site and extends to the right-hand side in a manner that is dependent on the number of consecutive windows placed along the alignment, as well as their width (Fig 2, step i); windows being regions from which nodes reflecting diversity are created. Both these parameters, as well as that of the clustering threshold that is applied within windows, can be altered using sliders located within the control area, thus allowing the user to visually explore the variation present. Once again there is a reset button to reset all network parameters to default values. Window locations are highlighted in orange along the bottom of the interface.

*(2.1) Nodes.* For a given window clustering fragments of sequences that span that window creates nodes (Fig 2, step ii). Each cluster is created using an iterative approach. Initially a fragment is randomly selected to be a seed for a newly created empty cluster. All related fragments to that seed are then added to the cluster and become seeds for the next iteration. The metric used to define relatedness is hamming distance, i.e. the number of different characters between

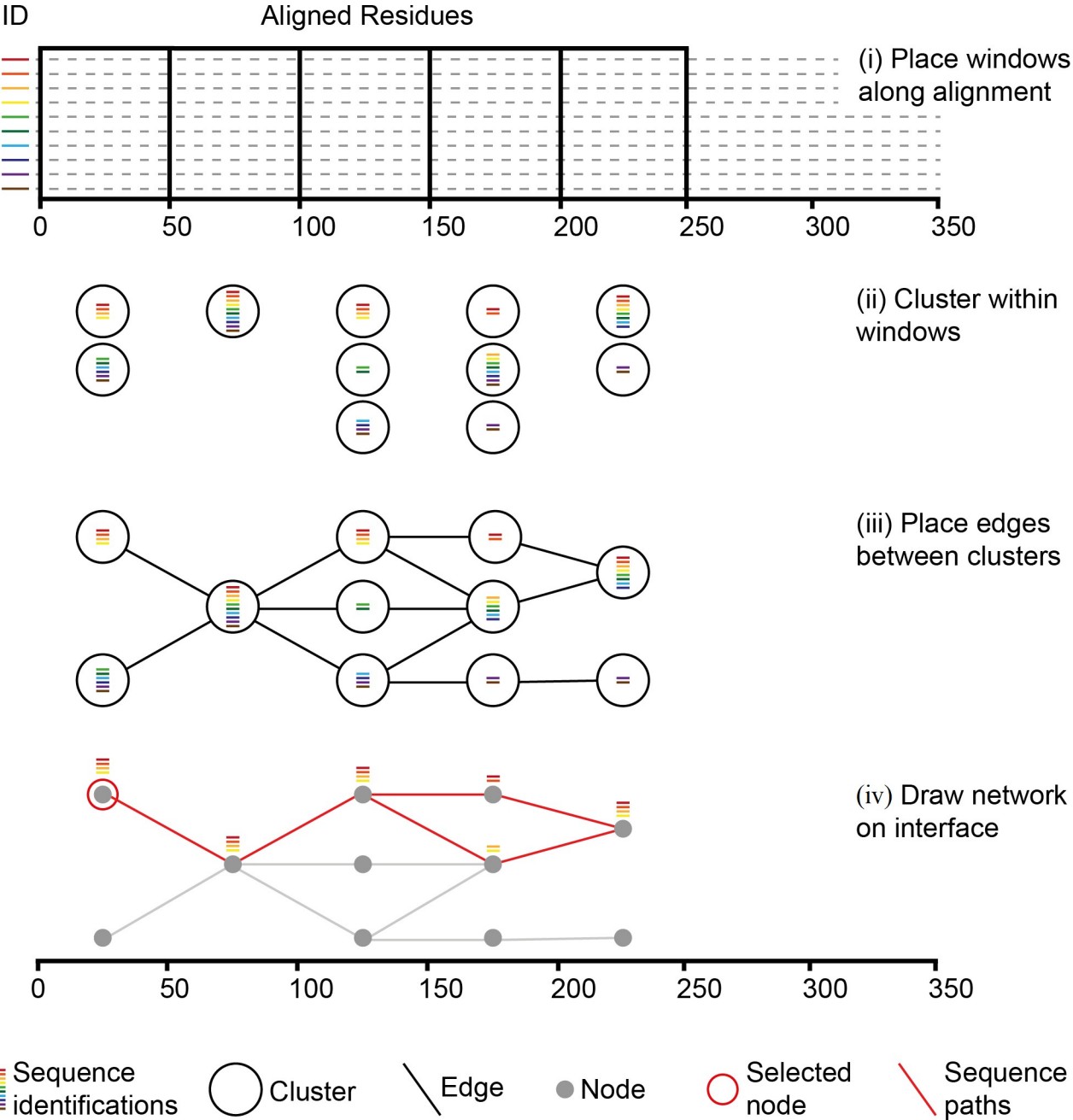

**Fig 2. Network construction.** Coloured bars indicate unique sequence id's relative to the corresponding sequences (dotted lines). Within each window the identification representing each full sequence are associated with individual sequence fragments spanning that window (i), and fragments within windows are clustered (ii). Edges are placed between neighbouring clusters where they share one or more sequence identification, i.e. differing regions of the same sequence (iii). Clusters are represented visually on the network by grey dots. If a single cluster is selected the paths of all sequences passing through in relation to all other clusters (red lines) can be traced (vi).

two aligned sequence fragments are counted. The default threshold value is a lenient 0.3, indicating that fragments that have less than 30% divergence from a seed are included within the cluster. More advanced measures of genetic distance exist that account for proposed models of sequence evolution at both nucleotide and amino acid levels [40, 41], but for the rapid clustering across windows placed along an alignment for the purpose of visualization hamming

distance works well [42]. Iterations continue until no more next-round seeds can be identified. If unclustered fragments within the window still exist, a new cluster is initiated by selecting another random seed from the remaining fragments and the process is repeated.

For windows where six or less clusters are created, all clusters are displayed as grey circles, or nodes, on the network. For windows with more than six clusters, the largest six are displayed as grey circles, whilst sequences from any remaining smaller clusters are placed into a holding structure that is used for visualization purpose and that is displayed as a black circle. This holding structure is treated in the same as any other node when it comes to tracking sequences that are within it, i.e. if it is clicked then the paths of all sequences passing through it will be highlighted across the rest of the network. Here six was chosen to be the upper display limit so that following edge placement (next section), and associated edge crossover minimization, the maximum number of nodes that need order re-arrangement within any one window is seven, including the holding node (if present). This is because for a set containing n items, there are n factorial different order permutations [43], and during edge crossover minimization the number of edge crossovers produced by each permutation, in relative to nodes within a neighbouring window, must be counted. For a given window if there are the maximum of seven nodes present, 7! permutations (5040) must be identified during crossover minimization and this can be done in a reasonable time ($< 1$ second on an average laptop). If on the other hand there are fifteen nodes allowed within a window, then there are 15! permutations (1307674368000) requiring a time of many days. We arbitrarily felt that six clusters, and the additional holding node, was a sufficient number to graphically indicate the main variant groups present whilst minimizing edge crossover calculation time; the latter maintaining parameter updates in real time for the user as they scroll through the alignment. However, this limitation is for interface visualization purposes only and a higher resolution of all clusters defined by up to a 1% divergence threshold is possible using the "All Sequences" -> "Cluster Sequences" option. The default number (10) and width (50 bp) of windows, as well as the pairwise distance threshold, can be altered using the slider bars within the "Navigation and Control" area. The number of sequences passing through each node is indicated in green on the left and right hand sides of the network, as well as within the hint box of the control panel area if a particular node is clicked on.

*(2.2) Edges.* Edges are placed between nodes of neighbouring windows where they possess fragments that are derived from the same underlying sequence(s) (Fig 2, iii). Consequently, individual sequences can be traced through nodes across different windows. As indicate in the previous section, edge crossover between nodes within adjacent windows is minimized. Starting at the second most right-hand-side window, this is done by calculating all possible node order permutations, following which for each permutation, the number of edge crossovers to nodes within the adjacent right-hand window is counted, node layout order in the latter being kept constant (Fig 3). The permutations that produce the minimum number of crossovers are selected (Fig 3, red numbers), and from these a random one is used. The process is then repeated one window to the left, until the first window of the alignment is reached. Crossover minimization, whilst resulting in a visually more pleasing network, has no effect on the sequence information or underlying node connections. Following the connection of edges it is possible to click on nodes within the graph and track the sequences that pass through them (Fig 2, iv). On the interface, and as described in detail within the "Sequence View" section, such sequence paths are displayed in red, and a red dot is placed next to the titles of the sequences within the clicked on node.

**(3) Navigation and control.** Access to all the network and display parameters is provided through the slider bars associated with the navigation and control panel (Fig 1), and in each case reset to default buttons have been provided. The buttons labelled with then red directional

## (i) Label clusters and identify required edges using shared sequence id´s between clusters

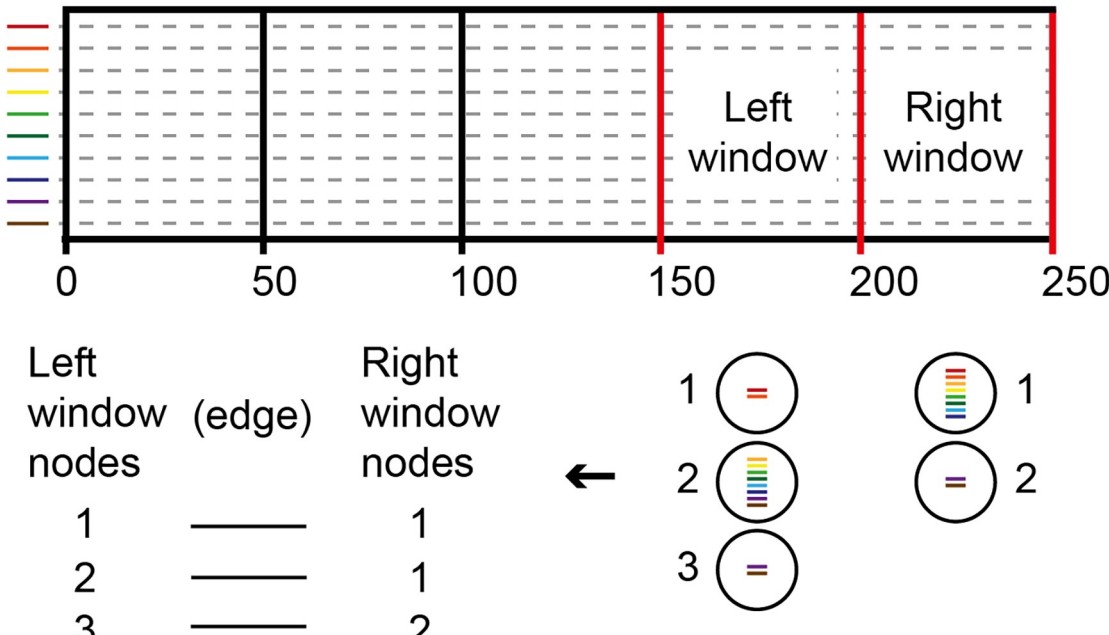

## (ii) Calculate left-hand-side node order permutations and corresponding edge crossovers

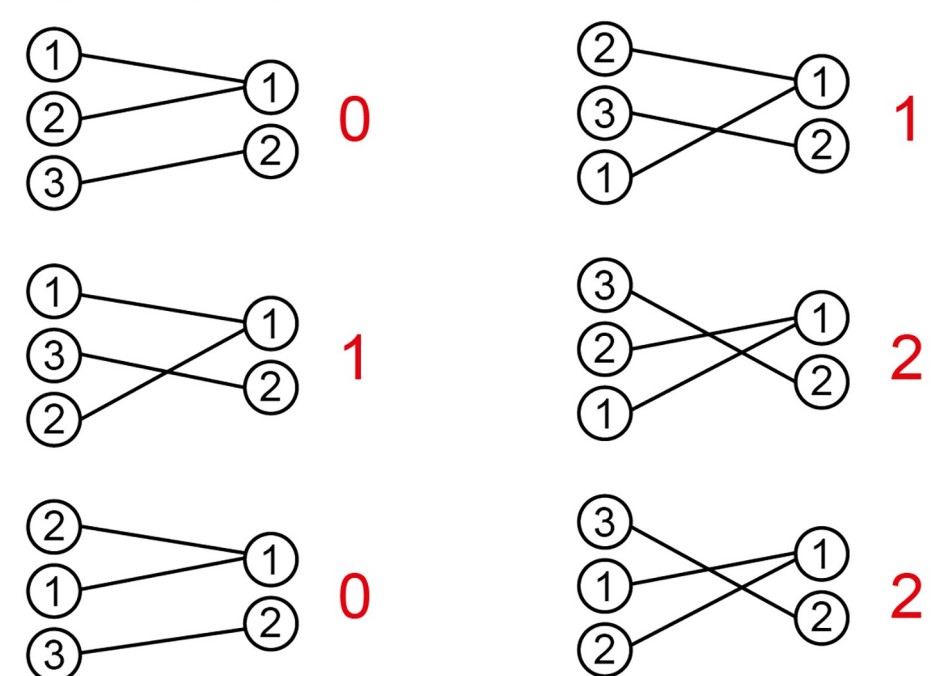

**Fig 3. Minimization of edge crossovers between nodes of the two right most windows of the alignment.** This process is repeated until the left most window (anchored on site 1) is reached. Clusters within the two windows are labelled with integers and required edges, based on the sequence ids (coloured bars), are listed (i). All order permutations of the current left window are identified and for each permutation the required edges are placed relative to the constant cluster order of the right window (ii). Crossovers are then counted (red numbers). Of the permutations that produce the minimum number of crossovers a random one is selected for graphical node layout order.

arrows are used to scroll through the alignment. These were implemented to remove the need for flat scroll bars as future developments will be aimed more at tablets and mobile devices. The red dot, at the centre of the four scroll arrows, immediately jumps a viewpoint at the centre of the alignment. In addition to the directional arrows the user can click directly on the grey squares along the alignment location indicator bar to immediately move to a particular location. Within this control area there are also buttons associated with for printing the network to a.png formatted file and clearing highlighted paths as well as a hint text section. The latter displays hint text from other areas of the display that the user clicks on such as sequences, nodes, location bars and general areas.

(4) **Menu driven output.** Output options are accessed through the top-level menu bar and can be applied to (i) all sequences within the alignment, (ii) a subset of sequences whose titles match a user search criteria, (iii) a subset of sequences that pass through a selected node and (iv) a subset of sequences defined by the user based on a supplied file of titles. Interacting with sequences through these predefined options increases output data robustness; as opposed to allowing the random the inclusion of sequences through mouse selection into the subsets that the various sub-menu options are applied to. In addition to exporting subsets of sequences and/or specified regions of the alignment CView can generate summary statistics such as frequencies of residues and kmers as well as tertiary information such as pairwise distance matrix's, variant count information and clustered sets of sequences. A description of sub-menu options is available on the wiki associated with the SourceForge project page, within the software itself under the menu option "Help" -> "Menu" (also accessible via the help button within the "navigation and control" area) and is outlined within the demonstration video [35].

## Results

### (1) The software

CView has been implemented in Java and runs on operating systems with installed Java Runtime Environment 8.0 or higher. It has been developed using an object-orientated approach for ease of plug-in development; where plug-ins related to alignment visualization will be based on user feedback, and placed under the "Plug-ins" menu. To obtain an executable jar file, download the cview.zip file from the SourceForge project page. Following the extraction the CView.jar file from the zip file, CView is executed by double clicking on the jar file. This will launch the interface through which alignments can be loaded. Alignments must be in fasta format where a '-' character usually represents a gap (occasionally a '.' can be used), whilst an 'N' character represents an ambiguously sequenced residue. Gap characters are inserted by the alignment software, e.g. MUSCLE [1] and Clustal W [2], whilst ambiguous characters are added during the sequencing process and usually in accordance to the IUPAC codes [44, 45]. Importantly, following the multiple alignment process (not performed by CView), sequences are all of the same length, and this is a requirement of the input of CView. Aligned fasta formatted sequences are loaded using the "Load Fasta (Alignment)" option of the "All Sequences" menu.

Future developments will aim to include other more data rich alignment formats, such as the Stockholm format developed by the Pfam [46] and Rfam [47] consortiums, but fasta was selected for the initial release version as it is widely adopted and allowed the development to focus on the display of aligned residues, the associated network and associated export features, rather than on the interactions involving metadata such as that involving secondary structure, surface accessibility, intron information and at times phylogenetic relationships. We view the latter as being more suitable to future plug-in's development designed to target specific data analysis pipelines that may be of less widespread interest. Once a fasta-formatted alignment has been loaded the workflow is driven by how the user interacts with the interface and the various output options. A test dataset, in the form of an alignment consisting of 636 sequences representing the gp120 region of the HIV-1 genome is included with the cview.zip download file. This data was obtained from the Los Alamos HIV sequence database [37] and is intended for initial testing of the software, but it is not the more complex alignment that was prepared during the test case example that follows.

## (2) Test case example: Exploring variation associated with co-receptor usage

**(2.1) Background.**  HIV-1 viruses can be characterized into two phenotypes that are dependent on cellular tropism and that are as a result of differences in co-receptor usage [48]. The macrophage tropic phenotype, often referred to as R5, requires the CCR5 co-receptor, whilst the T-cell tropic phenotype (X4) uses the CXCR4 co-receptor, the latter often emerging later on during infection [49]. Co-receptor usage can be detected by computational analysis based on specific genetic alterations within the V3 loop of the gp120 gene [29, 30]. Genetic variation within this region, of approximately 105 nt in length, lead to structural shifts that result in optimized binding to one co-receptor or the other [50]. For demonstrating the applicability of CView in aiding the characterization of sequence diversity, we have prepared a number of easy-to-follow steps involving the preparation and alignment of HIV-1 subtype B gp120 sequences representing each of the phenotypes described. Towards the end of these steps the V3 region of the alignment is extracted and per-site sequence variation between the two phenotypes is summarized. In this test case scenario the exact co-ordinates of the V3 loop are known prior to analysis, as are the phenotypes of each of the sequences present. However, this will often not be the case and a more exploratory approach could be adopted as indicated following the steps below.

**(2.2) Method.**

1. All North American subtype B gp120 sequences, verified to be CCR5-using sequences (n = 636), were downloaded from the Los Alamos HIV sequence database in aligned fasta format [37]. These were loaded into CView, using the "All Sequences" -> "Load Fasta (Alignment)" menu option, following which they were saved in unaligned format using the "All Sequences" -> "Save Unaligned" option. Additionally, the titles of these sequences were saved to a separate file using "All Sequences" -> "Save Titles".

2. Step 1 was repeated for CXCR4-using sequences (n = 76).

3. In order to make sites directly comparable between the two sets of unaligned sequences, they were combined into a single file (by simply copying and pasting using a text editor) and aligned using MUSCLE [1]. Note: steps 1 to 3 were performed in order to ensure the quality of the final alignment. Alignment containing both CCR5- and CXCR4-using sequences could have been downloaded directly from the database, but these will have been

extracted from a larger alignment containing many more sequences and so may not have been optimized.

4. The alignment created by MUSCLE, available at [51], was loaded into CView and the consensus sequence of the region spanning the V3 loop was saved using the "All Sequences" -> "Consensus Sequence" option. Within this alignment the co-ordinates of the region spanning the V3 loop were from 1436 to 1568. Although the exact location of the V3 loop within the gp120 region is known relative to the HXB2-LAI-IIIB-BRU reference strain (accession: K03455), the coordinates will vary depending on the alignment due to the placement of gap characters ('-') introduced during the alignment process. The exact co-ordinates of the V3 region within our alignment were identified by eye using the V3 sequence of the HIV-1 reference strain where the start residues of the loop are TGTACAAGACCC and the end residues are CAAGCACATTGT [37].

5. The proportion of the alignment corresponding to R5 sequences was saved to a separate file in aligned format; and labelled accordingly with R5 in the title. This was done using the "User Group" -> "Save Region" menu option, where the titles used to define the R5 group of sequences to be saved were those obtained during step 1. The latter was used as in the case of this data the sequence titles were not labelled with any motif/tag that could have been used to identify them using the "Search Titles" top-level menu option.

6. Step 5 was repeated for the titles corresponding to the X4 sequences (obtained during step 2) and the output file was labelled accordingly.

7. Steps 5 and 6 resulted in two sub-alignments, contained within separate files, where although they represented different phenotypes (R5 and X4) the per-site co-ordinates were compatible as they represented sequences from a common underlying alignment. Each of these two files were loaded into CView, and nucleotide frequencies spanning the V3 loop were obtained using the "All Sequences" -> "Residue Frequencies" menu option. The co-ordinates entered for the V3 region within the subsequent popup box were those used in step 3. Once again each output file was labelled accordingly, for example the frequencies obtained from the R5 alignment were labelled R5_frequencies.txt.

8. For each of the extracted R5 and X4 alignments obtained in steps 5 and 6, CView was also used to output a list of variants spanning the V3 loop along with their frequency of occurrence. This was done by loading each sub-alignment into CView and applying the "Variant Frequencies"option of the "All Sequences" menu and using the co-ordinates specified above.

9. Each of the two output files containing variants were then translated using the EMBOSS Transeq tool [52].

The underlying alignment described for this use-case scenario, consisting of all the HIV-1 SUBTYPE B sequences spanning the gp120 region of the genome that have been verified as either being a CCR5-using (n = 636) or CXCR-using (n = 76), is available from the zenodo repository [51]. Within this alignment the exact co-ordinates of the region of the gene that harboured the diversity associated with either the X4 or the R5 phenotypes were known prior to the analysis. However, for many alignments a more exploratory approach may be preferable. One way that this can be achieved is by toggling the number of windows, their size and the clustering threshold used in order to obtain a more global view of the diversity present, and by subsequently using the "Node Path" top-level menu option to extract portions of sequences, and/or information from sequences such as kmer frequencies, that pass though nodes on the network that are of visual interest.

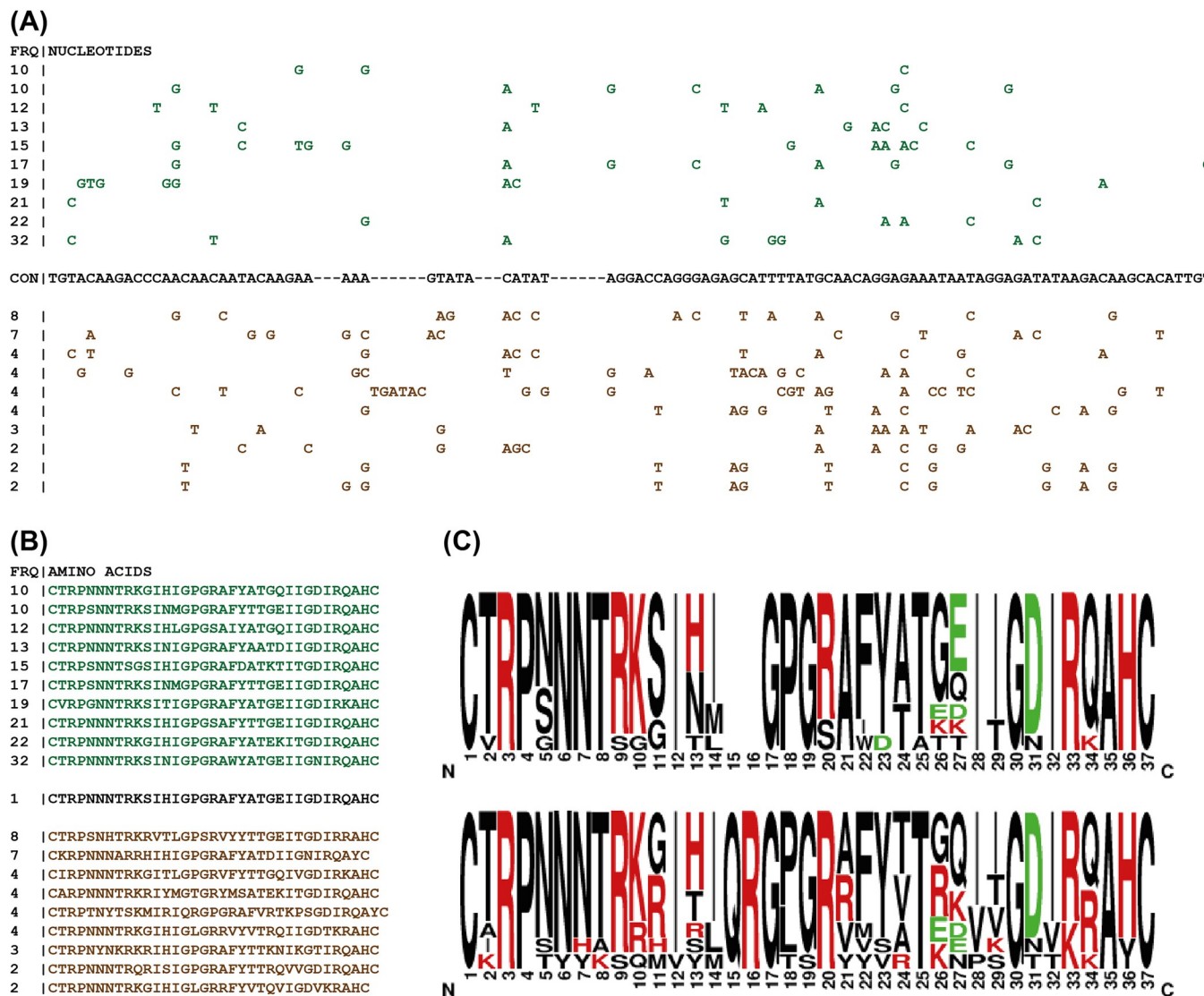

**Fig 4. Summarization of variation present within the V3 loop.** (A) Green residues represent non-consensus residues from the ten most frequent variants associate with the CCR5-using phenotype. Brown represents those of the CXCR4-using phenotype. The consensus sequence (black) is shown. (B) Translations of the most frequent ten variants from each phenotype. (C) Sequence logos summarizing these translations. The top logo is from represents the CCR5-using sequences whilst the bottom represents the CXCR4-using ones.

**(2.3) Result and discussion.** Fig 4A displays the consensus sequence of the V3 region from the MUSCLE generated alignment prior to being divided by phenotype. The top ten most frequent variants from each of the two phenotypes are also displayed. The seqPublish tool [37], located at https://www.hiv.lanl.gov/content/sequence/SeqPublish/seqpublish.html, was used to format these alignments from the CView output such that characters identical to those of the consensus sequence were hidden. A similar text-based formatting is available at the bottom of the output file that is generated by the "Variant Frequency" option of CView, where residues that are identical to those present of the most frequent variant are represented by a "|" character. Here we used the seqPublish tool as variants from both phenotypes were being compared to the combined consensus and not the most frequent variant from a single alignment representing a single phenotype. The translation of each of these variants is

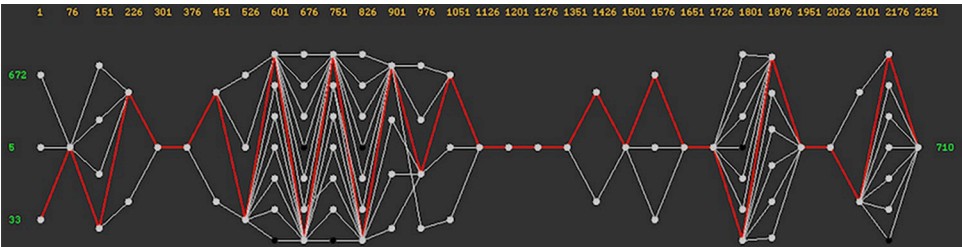

**Fig 5. Network based diversity across the full gp120 case-study alignment: Thirty non-overlapping windows of length 75 nt fitted along the length of the alignment.** The clustering threshold within each was 0.2. The orange numbers along the top indicate window locations, whilst the green numbers on each side indicate the number of sequences observed within nodes (clusters) at the ends of the network. Grey dots are clickable nodes, that when clicked will highlight the paths (in red) of the sequences passing through across the remainder of the network. Here the highlighted paths are a result of clicking the bottom left most node (containing 33 sequences). Information from sequences highlighted in this manner can be extracted using the sub-options of the "Node Path" top-level menu within the software as indicated within the demonstration video [35], and under the software help menus. This network was printed using the "Print PNG" button located within the control panel of the software.

presented within Fig 4B. Fig 4C shows a summary of the translations, obtained using sequence logos [53], and it can be observed that at site 11 the positively charged amino acid residues R (arginine) and H (histidine) are present within the sequences that were known to be CXCR4-using, whilst they are absent within the sequences obtained from the CCR5-uisng strains. At site 26 a similar observation is made in relation to positively charged residues, this time including a K (lysine) residue; although there is a minority K also present at 26 within the CCR5-using variants. This is a known observation where the presence of positively charged amino acids at sites 11 and 26 result in a structural alteration that optimizes CXCR4 co-receptor binding [29, 30]. The defined steps outlined in the use-case scenario presented here led to the summarization of the variation present within the V3 loop partially demonstrate the utility of CView when exploring aligned sequence data. The correctness of the summarization is confirmed by the presence of the known variation seen at sites 11 and 26. Complete per-site nucleotide frequencies for both R5 and X4 sequences spanning the V3 region are presented in supplementary S1 Table.

In the absence of pre-defined phenotypes, site co-ordinates and subtype, a more exploratory approach could be preferred. Fig 5 displays a picture of the network diversity observed across the entire gp120 alignment were the number and size of the windows have been maximized (to 50 and 75 respectively) and the clustering threshold set to 0.20. The user can click on individual nodes within the network, in order to identify subsets of sequences passing through, in order to export a range of information types that are implemented within CView for subsequent analysis relative to those subsets.

## Conclusion

CView is a tool that allows the user to interactively explore sequence alignments with the aid of a dynamic network that summarizes the diversity present within regions of the alignment not currently in-view. Here we have described how CView was designed and implemented as well as summarized the various output features available to the user. As a use-case example, we have characterized the variation within sequences involved in HIV-1 co-receptor usage, but we have also hinted at how CView can be used in a more exploratory manner to explore aligned sequence data. The exact usage scenario in which CView can be applied is dependent on the requirements, insight and background of the individual user.

## Supporting information

**S1 Table. Nucleotide frequencies from the V3 loop.** Sites covering codon 11 and 26 are highlighted in red.
(DOCX)

## Author Contributions

**Conceptualization:** John Archer.

**Data curation:** John Archer.

**Funding acquisition:** John Archer.

**Investigation:** John Archer.

**Methodology:** John Archer.

**Project administration:** John Archer.

**Resources:** John Archer.

**Software:** John Archer.

**Supervision:** John Archer.

**Validation:** Raquel Linheiro, Stephen Sabatino, Diana Lobo, John Archer.

**Visualization:** Raquel Linheiro, Stephen Sabatino, John Archer.

**Writing – original draft:** John Archer.

**Writing – review & editing:** Raquel Linheiro, Stephen Sabatino, Diana Lobo, John Archer.

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
