## [Decision Letter · Decision Letter 0]

21 Mar 2022

PONE-D-21-33624CView: A network based tool for enhanced alignment visualizationPLOS ONE

Dear Dr. Archer,

Thank you for submitting your manuscript to PLOS ONE. After careful consideration, we feel that it has merit but does not fully meet PLOS ONE’s publication criteria as it currently stands. Therefore, we invite you to submit a revised version of the manuscript that addresses the points raised during the review process. Specially take into account the comments related to fasta pre-processing, visualisation tips and configuration paramenters. 

We look forward to receiving your revised manuscript.

Kind regards,

Eduardo Andrés-León

Academic Editor

PLOS ONE

Reviewers' comments:

Reviewer's Responses to Questions

**Comments to the Author**

1. Is the manuscript technically sound, and do the data support the conclusions?

Reviewer #1: Partly

Reviewer #2: Yes

2. Has the statistical analysis been performed appropriately and rigorously? 

Reviewer #1: N/A

Reviewer #2: N/A

3. Have the authors made all data underlying the findings in their manuscript fully available?

Reviewer #1: Yes

Reviewer #2: Yes

4. Is the manuscript presented in an intelligible fashion and written in standard English?

Reviewer #1: Yes

Reviewer #2: Yes

5. Review Comments to the Author

Reviewer #1: The manuscript describes CView, a new software for visualising multiple sequence alignments using a graph-based approach. The text is easy to follow and I was able to install and run the software. However, I am not sure I fully understand how to use CView to its full potential, and although I agree with the authors that a new software such as CView could be useful to understand complex alignments, without extensive documentation and tutorials the users are unlikely to adopt the tool for their work.

Major comments

I struggled with the interface and found that the software requires additional context-specific documentation in the user interface, such as tooltips or popups. For example, the manuscript explains that a black circle is a holding structure that is used for visualisation purposes only when there are more than six clusters in the window. However, the user interface does not seem to explain why some nodes look like black circles and what is the meaning behind it. There are other examples along the same lines, for example, the red circles or the significance of colours in the network chart. These should be better explained in the program itself so that the users do not have to constantly refer to the paper to use CView.

Have the authors considered providing a video walkthrough or a tutorial for CView? As this is a new concept, it would be very useful to see an expert use the software, as I am not sure I am able to take full advantage of it having read the paper and the help section of the website.

I may have missed it, but as far as I can tell the use case does not take advantage of the network view, which I found surprising given that the graph aspect is the key strength of the method.

Minor comments

It would be useful to support importing alignments in Stockholm format to enable direct import from major alignment databases like Pfam and Rfam.

Would it be possible to add an option to restore defaults for all parameters? After tinkering with the sliders I quickly lost track of what the recommended values were.

Have the authors considered performing user testing to see how novice users or those experienced with other software (for example Jalview) interact with CView?

Reviewer #2: This article provides CView, a software for visualization of multiple sequence alignments (DNA/RNA and amino-acids). This software application favors small sequences, such as viral genomes or proteomes. The novelty is centered on the network analysis and additional components that interactively allow an easier data summarization.

The tool provides variant summarization, per-site character and kmer frequency matrixes, clustered sequences, pairwise-distance matrixes, and consensus sequence generation. The visualization is divided into four panels, the sequence view, network view, navigation and control, and menu-driven options.

Generally, the article is well written.

Regarding the software, I have successfully "installed" and tested it on a Linux machine using the HIV viral sequences example.

Then, I went to the NCBI and downloaded the B19V sequences in FASTA format. When I loaded these sequences into the CView, the following error emerged: "Exception occurred in newAlignment()". Then, I looked into the sequences and saw that the example had "-" instead of "N" symbols (which exist in some FASTA files but do not in the majority of the FASTA files). I changed the "N" to "-" using a Linux tr command and got the same error. Finally, I've tried it with other sequences and got the same error. Therefore, further than this action, I've not been able to test the software. This error is an obvious problem that must be fixed before any deeper review.

Minor subjects:

Once the bugs are fixed, please consider adding this software to bioconda and biootools;

Please, state the license type in the manuscript;

It would be nice to select some of the sequences by "mouse selection" to perform further actions;

Please, consider stating the existence of alignment-free methodologies (for example, to visual applications with larger viral genomes, such as Herpesvirus) and Galaxy;

Please, state clearly in the main manuscript (perhaps in the abstract) if this tool is open-source or not.

6. PLOS authors have the option to publish the peer review history of their article (what does this mean?). If published, this will include your full peer review and any attached files.

Reviewer #1: No

Reviewer #2: No

---

## [Author Response · Author response to Decision Letter 0]

5 May 2022

The text we here is the same as that within the uploaded response_to_reviewers.docx file. We have included the text here as this box was compulsory on the submission form.

Reviewer #1 

Summary

"The manuscript describes CView, a new software for visualising multiple sequence alignments using a graph-based approach. The text is easy to follow and I was able to install and run the software. However, I am not sure I fully understand how to use CView to its full potential, and although I agree with the authors that a new software such as CView could be useful to understand complex alignments, without extensive documentation and tutorials the users are unlikely to adopt the tool for their work."

We thank this reviewer for downloading, running and reviewing our software and manuscript. Additionally, we are pleased that the reviewer sees the scope that new alignment visualization tools could be of value in understanding sequence data. We agree that further documentation is required for users to widely adopt our tool and as such we have worked extensively on this and the other recommendations as outlined below.

Major comments

"I struggled with the interface and found that the software requires additional context-specific documentation in the user interface, such as tooltips or popups. For example, the manuscript explains that a black circle is a holding structure that is used for visualisation purposes only when there are more than six clusters in the window. However, the user interface does not seem to explain why some nodes look like black circles and what is the meaning behind it. There are other examples along the same lines, for example, the red circles or the significance of colours in the network chart. These should be better explained in the program itself so that the users do not have to constantly refer to the paper to use CView."

We have added three in-software documentation sections under the “Help” top-level menu that can be accessed at any time during usage of the tool. These are: (1) “Help” -> “Interface”, contains brief graphical overviews of the software interface, the sequence area and the network area. (2) “Help” -> “Menu” contains textual descriptions of the methods that can be accessed via the other top-level menu options and (3) “Help” -> “Plug-ins” will contain information on the plug-ins that will be developed; so far there is one example titled “Variant Scan”. These in-software help menu options have been mentioned within the manuscript at lines 164, 208 and 391, as well as within the legend of figure 5 at line 597. Access to the first two help popup windows can also be achieved through the two new buttons within the navigation region of the control area (bottom left of display) as has been mentioned on lines 209 and 392.

Under the image associated with network area (“Help” -> “Interface”), black circles are labelled as containing sequences that are not within one of the largest six clusters (if more than six exist). The reason for this limitation of six of the largest clusters within any given window is due to reducing the time required to minimize edge crossovers for network visualization network and is described in detail within the manuscript (around line 299). We have also included a new figure 5 (line 597), describing a global view of the alignment used for the case study scenario, where these black circles are also mentioned. 

Within the in-software interface help image, under “help” -> “Interface”, it has been made clear that the red circles next to sequence titles belong to sequences that pass though a user clicked-on node, whose paths also are shown in red on the network. This is discussed in detail within the manuscript at line 238. The previous grey circles that were next to the sequence titles, if the red ones were not present, have been removed as they were likely to be adding confusion to the interface, and were not adding real information value. They were initially used as placeholder indicating that the individual sequence was not passing through a user-clicked node. 

The interface has been modified so that when an area is clicked on, a hint is displayed within a newly added hint box at the top of the control area of the display. For example, if the user clicks on the ‘+’ symbols under the consensus sequence then these are explained as “'+' indicates residue conserved across all sequences”, whilst if a node is clicked on the location of the node and the number of sequences passing through it are given. If a black placeholder node (black circle) is present and clicked on, the hint text provided is “sequences not in top six clusters” and the total number of sequences within the holder is given. This hint text addition is mentioned at lines 210 and 327 and is also visible within figure 1 and under the “Help” -> ”Interface” in-software help image. Note: with the exception of standard sliders and buttons, we chose to display hint text once an item is “clicked on”, rather than “hovered on”. This is because this feature was implemented in a bespoke manner for our general display area and hover text would mean that the software must constantly monitor the cursor location when it is being moved. With a click this constant monitoring is avoided and the background monitoring that may at times affect the speed of updating the display could be avoided.

"Have the authors considered providing a video walkthrough or a tutorial for CView? As this is a new concept, it would be very useful to see an expert use the software, as I am not sure I am able to take full advantage of it having read the paper and the help section of the website."

We have created a walk through video of the basic usage of CView. This can be accessed through the citable zenodo repository link (https://zenodo.org/record/6514787), along with a text based outline of the script used within the video. A reference to this video tutorial has been provided at lines 164, 393 and 599, and is also hyperlinked from the CView SourceForge project page wiki (https://sourceforge.net/p/cview/wiki/Home/). Also included at line 165 as a separate zenodo reference is a small PDF based tutorial in relation to identifying variants using CView. Additional tutorials will follow as features are added to the software and as user scenarios are identified.

"I may have missed it, but as far as I can tell the use case does not take advantage of the network view, which I found surprising given that the graph aspect is the key strength of the method."

Choosing a test case scenario was difficult, as our aim was to develop a “basic”, or “lite”, alignment viewer that adds insight into the diversity observed within residues on-screen by looking at the context of regions off-screen. Additionally, we wanted to include an array of straightforward output features that other tools sometimes ignore; such as that of kmer information, variant counts and hamming distances. Our goal was not to come up with complex scenarios where CView excels over other viewers, but to attempt to enhance the basic experience of viewing and exploring sequence data. As it happens, if a user is looking at an alignment, where at some location there are two distinct clusters of diversity, then being able to click on one of these and immediately identify and extract sequences, and information within, “could” be of interest. But we did not start off with this being a main goal of functionality. Our main goal was to keep the interface as simple possible whilst augmenting intuitive insight and output, as implied throughout the introduction of manuscript e.g. line 128. 

Our current test case scenario, using HIV-1’s gp120 gene, comes from the angle were it is known that within the gene there is a small 105 nt region where residue alterations involving a few sites define which host co-receptor the virus can use. We felt that by being able to explicitly indicate the co-ordinates of this region from a set of longer aligned gp120 sequences, and extract information from sequences representing virus known use either one co-receptor or the other, it provided us with a way of demonstrating how the information that CView outputs, in this case that of variant frequencies and nucleotide frequencies, can be utilized to characterize diversity present when following a number of pre-defined steps. The other scenario, hinted at by the reviewer, is where the user may not know much detail about the alignment, but by “playing with” the network parameters identify clusters of similarity and then export information from sequences passing though these for further analysis. This is of interest and is indeed one way that CView can be used to explore alignments, hence why the “Node Path” top-level menu was added, but its requires a more subjective toggling of network parameters, and the steps are harder to define for a step-by-step demonstration of the simple and intuitive viewing and export features. 

We would like the reviewer to consider this and understand our choice in maintaining our HIV-1 gp120 example. That being said, we have specifically emphasised at the start (line 454) and end of the case study (line 529) that the user could adopt a more exploratory approach. We have elaborated on this slightly within the paragraph starting at line 578 where a new figure 5 (line 588) is used to highlight the discussion. In relation to the HIV-1 case study, we have clarified some of the steps outlined (from line 458) and placed the alignment generated along with the sequence titles within the zenodo repository where it is associated with a citable doi number (lines 477 and 522). Previously this data was available form the SourceForge project wiki, but we feel that it is more suited to a permanent citable repository and so has been moved from the software download page. Test data is still available within the cview.zip download file, but it has been made clear that this is not the case-study data (line 432).

Minor comments

"It would be useful to support importing alignments in Stockholm format to enable direct import from major alignment databases like Pfam and Rfam."

We agree that increasing the number of input data formats is important, but initially chose fasta format as it is widely used when dealing with aligned sequences, and no additional visualization features were required as there is no specific metadata fields (outside of what information is contained within the title lines). The ability to display metadata fields that can be associated with more advanced sequence input formats has not been incorporated and such information would immediately move away from the level of visualization simplicity we currently aimed at. 

We definitely agree that implementing more advanced formats, such as the Stockholm format, would achieve a wider usage of our tool, but we see this as advanced development post release since the current interface would need to incorporate the ability to display such additional metadata appropriately. For example, in the Stockholm format, some of the extra data allowed for RNA sequences are fields describing Secondary Structure, Surface Accessibility, TransMembrane, Posterior LIgand binding, Intron information as well as possible trees. Including such information within groups of aligned sequences whilst creating a network representing the alignment is a significant added complication to the interface, and we feel that this should be achieved though a specifically designed plug-in to accommodate users interested in such fields. A half-way point could be to simply ignore the extra metadata, effectively only using the titles and sequence residue information, but this would be misleading in terms of the initial functionality of our software; as it would not really be utilizing the richness of the Stockholm format, whilst at the same time portraying that it can load such data. 

Given that: (i) this is classified as a minor comment by the reviewer, (ii) our tool has been initially designed for the simple intuitive viewing of alignments and (iii) we do have a plug-in menu where we intent to develop more user specific requirements, we would request that the reviewer can overlook this limitation; and see the current input option as a base for further more advanced developments incorporating metadata specific fields. In relation to the reviewers comment we have extensively discussed the Stockholm format within the manuscript as well as the reason behind the current input limitation (starting at line 417).

"Would it be possible to add an option to restore defaults for all parameters? After tinkering with the sliders I quickly lost track of what the recommended values were."

Yes, we have added two buttons to the control area. One of these resets the network parameters to default, whilst the other resets the general display parameters to defaults. These buttons have been mentioned within the main manuscript at lines 195, 250, 265 and 363. These reset buttons are also visible in figure 1 and under the in-software interface help figure located under the “Help” -> “Interface” menu option.

"Have the authors considered performing user testing to see how novice users or those experienced with other software (for example Jalview) interact with CView?"

At the moment we have not, although we see this as a valuable approach. Currently during development we have utilized our personal backgrounds in sequence viewing to estimate what users could be interested in. Our main concern was to increase the intuition that can be gained when viewing an alignment. Once we have established our tool and its usage within a wider set of users, we would be pleased to perform user-base comparisons on how users interact with differing tools and include components of such interactions into our future developments of the interface. This would include, for example, users interested in specific areas of research such as metagenomics, viral genomics, phylogeny, recombination detection analysis and sequence classification as well as increase the output connectivity to external software. For example, networks could also be exported in a format that can be loaded into cytoscape, a tool that can be used to enhance the visualization of networks, add additional metadata to individual nodes and elaborate on the number of layout options. We see this as future tuning and user guided developments of our basic release interface once the initial concept has been established.

Reviewer #2

Summary

"This article provides CView, a software for visualization of multiple sequence alignments (DNA/RNA and amino-acids). This software application favors small sequences, such as viral genomes or proteomes. The novelty is centered on the network analysis and additional components that interactively allow an easier data summarization.

The tool provides variant summarization, per-site character and kmer frequency matrixes, clustered sequences, pairwise-distance matrixes, and consensus sequence generation. The visualization is divided into four panels, the sequence view, network view, navigation and control, and menu-driven options."

We thank this reviewer for taking the time to review our software and we have responded to the comments below. The comments provided helped us greatly to clarify our manuscript.

Major comments

"Generally, the article is well written. Regarding the software, I have successfully "installed" and tested it on a Linux machine using the HIV viral sequences example. Then, I went to the NCBI and downloaded the B19V sequences in FASTA format. When I loaded these sequences into the CView, the following error emerged: "Exception occurred in newAlignment()". Then, I looked into the sequences and saw that the example had "-" instead of "N" symbols (which exist in some FASTA files but do not in the majority of the FASTA files). I changed the "N" to "-" using a Linux tr command and got the same error. Finally, I've tried it with other sequences and got the same error. Therefore, further than this action, I've not been able to test the software. This error is an obvious problem that must be fixed before any deeper review."

Usually within aligned fasta formatted sequences a ‘-’ character represents a gap (occasionally a ‘.’ can be used), whilst an ‘N’ character represents an ambiguously sequenced residue. Gap characters are inserted by the alignment software (such as MUSCLE), whilst ambiguous characters are added during the sequencing process, and usually in accordance to the IUPAC codes. Importantly, sequences aligned by a software, such as MUSCLE, are all of the same length and this is a requirement of the input of CView. We have extensively clarified this definition of what we mean by aligned fasta format (for input) on line 406 to 415 and supplied a reference referring to the IUPAC codes near this clarification. Note: we specifically did not refer to our tool as a sequence viewer, or editor, as it is only for viewing pre-aligned sets of sequences that adhere to the input restrictions mentioned above. Viewing unaligned sequences is not possible, unless by chance they are all of the same length, but even here the network and output options will not function as intended.

Additionally, within the software itself we have added a warning message that indicates that “Sequences must be aligned and in fasta format. Please see the 'Help' -> 'Menu' option above.” rather than output the exception that the reviewer previously observed. Within the “Help” -> “Menu” option, we have also described this under the title “(i) Load Fasta” and recommended both Muscle and Clustal as example tools for creating an alignment; although many others exist. This is also mentioned within the demonstration video, and corresponding PDF file, available from the zenodo repository (https://zenodo.org/record/6514787) as referenced within the manuscript on lines 164, 393 and 599 (and is hyperlinked from the CView SourceForge project page wiki).

Minor comments

"Once the bugs are fixed, please consider adding this software to bioconda and biootools; Please, state the license type in the manuscript; It would be nice to select some of the sequences by "mouse selection" to perform further actions;"

The license agreement (GNU General Public License) has now been stated clearly in the manuscript at line 77 of the abstract and at line 199 of the introduction. At line 75 of the abstract it is now clear that the software is open source.

We have provided the “Title Search” top level menu option where the user can interact with groups of sequences through a user specified search criteria, such as a year or subtype, that will be searched for within the sequence titles. Additionally, we have provided the “User Group” top-level menu options where the user can interact with groups of sequences defined by a user specified set of titles, supplied as a text list. An example of supplying such a set of titles is given within the pdf script that accompanies the demonstration video (cited on lines 164, 393 and 599). We have clarified this functionality further with the user documentation as well as within the manuscript (line 382). 

Selecting multiple sequences using the mouse has not been implemented as we want to encourage the user to explore the alignment through interaction with the associated network through defined groups such as those specified above or through network nodes. For example, if the user clicks on a node on the network, then within the sequence view area all sequences passing through that node are highlighted with a red dot and are arranged to the top of the sequence list (discussed at line 238). Methods can then be applied to these through the “Node Path” top-level menu option. If the user can drag the mouse and select random sequences at will then such sequences will likely pass though multiple nodes on the network and no visual intuition will be achieved. We feel that interacting with sequences through the “Title Search”, “User Group” and “Node Path” menus options increases the definition of what information is contained with the various output options and improves output data robustness. This has been discussed within the manuscript starting at line 378.

In terms of considering to adding CView to bioconda and biootools, initially we were not sure how an alignment viewer would fit such platforms and this is why we created it as an open source sourceforge project. However, these platforms offer an interesting opportunity for us to increase our user-base and this is something we would like to look into further as the user scope of our tool becomes more clear.

"Please, consider stating the existence of alignment-free methodologies (for example, to visual applications with larger viral genomes, such as Herpesvirus) and Galaxy; Please, state clearly in the main manuscript (perhaps in the abstract) if this tool is open-source or not."

The existence of alignment free method has now been explicitly mentioned within the introduction, as has the reasoning we chose to create a tool based on aligned sequences. This inclusion is at line 112 and we have added some references in relation to it.

---

## [Decision Letter · Decision Letter 1]

31 May 2022

CView: A network based tool for enhanced alignment visualization

PONE-D-21-33624R1

Dear Dr. Archer,

We’re pleased to inform you that your manuscript has been judged scientifically suitable for publication and will be formally accepted for publication once it meets all outstanding technical requirements.

Kind regards,

Eduardo Andrés-León

Academic Editor

PLOS ONE

Additional Editor Comments (optional):

Reviewers' comments:

Reviewer's Responses to Questions

**Comments to the Author**

1. If the authors have adequately addressed your comments raised in a previous round of review and you feel that this manuscript is now acceptable for publication, you may indicate that here to bypass the “Comments to the Author” section, enter your conflict of interest statement in the “Confidential to Editor” section, and submit your "Accept" recommendation.

Reviewer #1: All comments have been addressed

Reviewer #2: All comments have been addressed

2. Is the manuscript technically sound, and do the data support the conclusions?

Reviewer #1: Yes

Reviewer #2: Yes

3. Has the statistical analysis been performed appropriately and rigorously? 

Reviewer #1: N/A

Reviewer #2: Yes

4. Have the authors made all data underlying the findings in their manuscript fully available?

Reviewer #1: Yes

Reviewer #2: Yes

5. Is the manuscript presented in an intelligible fashion and written in standard English?

Reviewer #1: Yes

Reviewer #2: Yes

6. Review Comments to the Author

Reviewer #1: I would like to thank the Authors for addressing my comments and including a video tutorial which I found very helpful. I was able to run the updated version of the software without any problems and I don't have any further comments. I look forward to the future versions of the software that would support the Stockholm format.

Reviewer #2: The authors have addressed my concerns. I'm happy to see that this software is open-source, a significant advantage over other software. Also, the video significantly improved the easiness of learning about this tool. Below are some points that could improve the manuscript and/or software.

Another advantage of using alignment methods is the enhanced local resolution. In my view, alignment-free and alignment methodologies are complementary.

Convincing a researcher to learn and use software intensely depends on multiple characteristics, including flexibility, simplicity, novelty, and communication. The video has dramatically improved the communication (Please, consider using Github for the next time - it offers much more features than Sourceforge; for example, the inclusion of a video online on the provided website, improved visualization of the repository, and custom characteristics). The simplicity and novelty are achieved with the good visualization usability and features it provides (great networking synchronized with the view of the alignment). The flexibility is perhaps the only feature that could contain an enhanced feature. Allowing CView to perform alignments would enable it to avoid multiple tools and provide a comprehensive offer. Please, consider in the future to add this option.

It would be great to have the option to change the default colors (perhaps in the main menu tab). This feature would enable a customized version of the software (also for the generation of the image prints) and please different individuals.

7. PLOS authors have the option to publish the peer review history of their article (what does this mean?). If published, this will include your full peer review and any attached files.

Reviewer #1: **Yes: **Anton Petrov

Reviewer #2: No

---

## [Editor Report · Acceptance letter]

2 Jun 2022

PONE-D-21-33624R1 

CView: A network based tool for enhanced alignment visualization 

Dear Dr. Archer:

I'm pleased to inform you that your manuscript has been deemed suitable for publication in PLOS ONE. Congratulations! Your manuscript is now with our production department. 

Kind regards, 

on behalf of

Dr. Eduardo Andrés-León 

Academic Editor

PLOS ONE